# The Impact of Anemia-Related Early Childhood Caries on Parents’ and Children’s Quality of Life

**DOI:** 10.3390/medicina59030521

**Published:** 2023-03-07

**Authors:** Dila Özyılkan, Özgür Tosun, Aylin İslam

**Affiliations:** 1Pediatric Dentistry Department, Faculty of Dentistry, Near East University, Nicosia 99138, Cyprus; 2Department of Biostatistics, Faculty of Medicine, Near East University, Nicosia 99138, Cyprus; 3Pediatric Dentistry Department, Faculty of Dentistry, European University of Lefke, Lefke 99010, Cyprus

**Keywords:** anemia, quality of life, children, parent

## Abstract

*Background and Objectives*: Today, oral diseases are well-known for their effects, not only on daily life but also on quality of life (QoL). Dental caries, especially early childhood caries (ECC), are considered a public health concern as regards their impact on the life quality of children and parents from multiple aspects. The present research was conducted to assess the effect of anemia on oral-health-related quality of life (OHRQoL) in terms of children and parents. *Materials and Methods*: The current study was performed in two independent stages. In the first stage, the Turkish version of the Early Childhood Oral Health Impact Scale (ECOHIS), and in the second stage, the Turkish version of the Parental-Caregivers Perceptions Questionnaire (P-CPQ) were used to measure the effect of anemia-related dental caries among children and parents. SPSS and Jamovi software were used for all calculations, graphs and comparisons. *Results*: A total of 204 participants (child–parent pairs) were incorporated in the present study. A considerable number of children (81.5%) reported occasional or more frequent oral/dental pain. Secondly, the subscale scores were determined for child symptoms (2.25 ± 0.067), child function (6.8 ± 0.22), child psychology (3.87 ± 0.128), self-image and social interaction (1.74 ± 0.063), parental distress (3.82 ± 0.143), and family function (3.5 ± 0.121). Additionally, more than half of the parents (56.3%) responded “fair” for the health of their children’s teeth, lips, jaws and mouth. Similarly, the child’s overall well-being was stated as being affected “*a lot*” by the condition of their child’s teeth, lips, jaws or mouth by half of the parents (49.5%). *Conclusions*: Anemia-related dental caries has a highly negative impact on the quality of life of children and parents according to both of the questionnaires. Therefore, children with high scores should be prioritized for preventive procedures and timely dental treatments.

## 1. Introduction

Today, oral diseases are well-known for their effects not only on daily life but also on quality of life (QoL), as oral health exhibits the main physiological, social and psychological characteristics that are fundamental for quality of life [1,2]. In this sense, the evaluation of oral health related to quality of life (OHRQoL) has come into prominence. The existence of dental disease, treatment experience, and oral health issues can cause unfavorable effects on the quality of daily life of children and parents [3,4,5]. Dental caries, especially early childhood caries (ECC), sre considered a public health concern due to their impact on the quality of life of children and parents through multiple aspects such as pain, eating disorders, sleeping problems, taking time off from school, social embarrassment for children, and financial problems related to treatment fees and time off work for parents [6,7,8]. Evaluating and measuring the child’s OHRQoL to help display the priority of care and the interpretation of treatment outcomes have gained popularity [9].

Even though various studies have mentioned the multifactorial etiology of ECC, including behavioral, socioeconomic, biological, and environmental factors, the relationship between anemia (particularly associated with malnutrition, iron deficiency, and vitamin D deficiency) and ECC has recently been highlighted. According to recent studies, it has been shown that there is a correlation between ECC and anemia, which is defined as the number of red blood cells or the oxygen-carrying capacity of blood being below normal levels [10,11,12,13]. Particularly, the possible mechanism of iron deficiency anemia (IDA) in the development of dental caries is the potential inhibitory effect of iron on cariogenic microorganisms [14]. The clinical and practical significance of iron and ferritin deficiency on dental caries was identified by the findings of a recent meta-analysis, which reported that children with caries had significantly lower levels of salivary and serum iron and ferritin [15]. Previous cross-sectional and case studies have also demonstrated that children with anemia or IDA have a higher risk of caries development. Additionally, the anticariogenic potential of iron in *Streptococcus mutans (S. mutans)* has been shown through the following mechanisms: (1) prevention of enamel demineralization in an acidic environment; (2) decreasing the level of dental plaque acidity, and (3) disinfectant and bacteriostatic capacity in *S. mutans*; and (4) inhibition of the glycosyltransferase activity [16,17,18,19,20,21]. Although the reasons regarding the theoretical basis for the link between anemia and ECC are described and discussed, there has been no research reported in the literature that underlines the effect of anemia-related dental caries on the OHRQoL of children and families. To date, numerous questionnaires have been developed in order to evaluate the OHRQoL for children, adults, and for families. However, self-reports of very young children on their oral health tend to be unreliable, and the reports are inefficient for different age groups [22]. Hence, the Early Childhood Oral Health Impact Scale (ECOHIS) was designed to measure the relationship between parents’ perceptions of their children’s quality of life and their oral health [23] together with the Parental-Caregivers Perceptions Questionnaire (P-CPQ), which is another scale that is used to evaluate oral health quality of the child from the perception of the parental/caregiver by providing good evidence of responsiveness [24].

In the current literature research, no study was found showing the effect of the presence of anemia on OHRQoL. The present research was conducted to assess the effect of anemia on oral-health-related quality of life (OHRQoL) in terms of children and parents.

## 2. Materials and Methods

### 2.1. Ethical Approval

The present cross-sectional study was conducted between June 2022 and August 2022 and performed at the Dr. Burhan Nalbantoğlu Government Hospital located in Northern Cyprus. The current research was approved by the Dr. Burhan Nalbantoğlu Government Hospital (YTK 1.01-30/22).

### 2.2. Data Collection and Participants

All data were collected following ethical approval. Firstly, all parents (mother/father) of children attending the Dr. Burhan Nalbantoğlu Government Hospital and diagnosed with anemia were analyzed and detected from the hospital’s archive. Later, the children aged between 2 and 18 years who presented at the dental clinic (Dr. Burhan Nalbantoğlu Government Hospital) with the complaint of dental caries, and without any accompanying systemic disease or an otherwise noncontributory medical history (except for anemia), were determined as meeting the inclusion criteria of the current study. Children with no anemia diagnosis or dental caries, parents living outside of Northern Cyprus, uncooperative children and families, children with a mental disorder, children who applied to the dental clinic with signs that constituted advanced infection (dental trauma/cellulitis), and children undergoing long-term antibiotic treatment were excluded from the study. Parents who enrolled in the study did so on a voluntary basis and received an informed-consent form to answer their questions.

The children who participated in the present study were screened by two of the pediatric dentists for the presence or absence of S-ECC, according to the AAPD guidelines [25], the first time they applied at the dental clinic. Five percent of all inspection objects were randomly selected and repeated for each day. Repeat checks were carried out in agreement with inter-investigator agreement.

### 2.3. Sample Size Determination and Questionnaire Design

G* power software (Ver.3.1.9.4) was used for the calculation of sample size. With a confidence level of 95% and a confidence interval of 7.5%, the targeted statistical power of 80% was calculated as 190. The current study was performed in two independent stages. In the first stage, the Turkish version of the ECOHIS was used, and in the second stage, the Turkish version of the P-CPQ was used to measure the effect of anemia-related dental caries among children and parents [26,27].

The questionnaire consisted of 61 questions divided into three parts: demographic information, ECOHIS questionnaire, and P-CPQ. Questions about parents’ relationship to the child, age, education level, occupation, marital status, number of children, income level, and their duties are asked in the demographic information section. Medication prescribed for the anemia, age of diagnosis, and age of starting medication questions were determined, and responses were recorded in the demographic information. The second part of the questionnaire includes the Early Childhood Oral Health Impact Scale (ECOHIS), which assesses Oral Health Related Quality of Life (OHRQoL). In the present study, the accepted and reliable Turkish form of the ECOHIS was used (26). The scale includes two parts, child impact and family impact, with 13 questions. The child impact part has subscales of child symptoms (1 question), child function (4 questions), child psychology (2 questions), and child self-image and social interaction (2 questions). The family impact part includes parental distress (2 questions) and family function (2 questions). Response classes for every question were rated on a five-point Likert scale: 0 = never; 1 = hardly ever; 2 = occasionally; 3 = often; 4 = very often; 5 = don’t know. For the child impact section, the score for children can vary from 0 to 36, and for the family impact section, the score can vary from 0 to 16, and so the system creates a total score range of 0 to 52. ECOHIS scores were calculated separately as a straightforward addition of the response codes for the child impact and family impact sections. According to the scale, higher scores demonstrate greater impact. For the ECOHIS scale, all “don’t know” (DK) responses were coded as missing. For the third part, the Turkish-accepted validity and reliability version of the Parental-Caregiver Perceptions Questionnaire (P-CPQ) was used to evaluate the QoL of children [20]. The scale includes 31 questions with 4 subscales of oral symptoms (6 questions), functional limitations (8 questions), emotional well-being (7 questions), and social well-being (10 questions). As with the ECOHIS, the P-CPQ was also rated on a 5-point Likert scale of never: 0; once/twice: 1; sometimes: 2; often: 3; and every day/almost every day: 4. General well-being was scored as follows: not at all = 0; very little = 1; somewhat = 2; a lot = 3; and very much = 4. For subscales, scores can vary for oral symptoms from 0–24, for functional limitations from 0–32, for emotional well-being from 0–28, and for social well-being from 0–40. Total scores were added, and a total-scale score was obtained. Similarly to the ECOHIS scale, a higher score represents a worse effect on quality of life. Parents who enrolled in the study did so on a voluntary basis and received the informed consent form of the study. Before the participating parents were selected, children were confirmed to have dental ECC. 

### 2.4. Statistical Analysis

Descriptive statistics for qualitative variables (frequency and percentage) and quantitative variables (arithmetic mean ± standard deviation) were calculated. Parametric test assumptions were controlled for the data, and the Shapiro–Wilk or Kolmogorov–Smirnov normality test was applied where appropriate. Non-parametric hypothesis tests were accordingly performed due to the distribution characteristics. For two independent groups, the Mann–Whitney U test was applied to evaluate the significance of the differences. In the case of more than two independent groups, the Kruskal–Wallis test was performed. In the case of significance, the Dwass–Steel–Critchlow–Fligner test was further performed to assess the pairwise significance. To investigate the associations between scale scores, the Pearson correlation test was performed. Cronbach’s alpha was calculated for each scale and subscale score to evaluate the reliability of the surveys. The level of significance was accepted to be 0.05. SPSS (Demo version 25.0 for Mac) and Jamovi (Version 2.2.0) software were used for all calculations, graphs, and comparisons.

## 3. Results

### 3.1. Sociodemographic Characteristics of Parents and Children

A total of 204 participants (child–parent pairs) were incorporated in the present study. Among the participants, 146 (71.6%) were mothers, and 58 (24.8%) were fathers, while 94 (46%) of children were male, and 110 (54%) were female. The age of parents ranged between 21 and 30 (26/12.7%), 31 and 40 (104/51%), and >40 (74/36.3%). Furthermore, the majority of respondents (79.3%) were well-educated (graduate–postgraduate level). Anemia type and medical treatment for this disease were also assessed. Responses related to anemia type were as follows: 54.5% for iron deficiency anemia, 29.5% for 25-hydroxyvitamin D deficiency anemia, and 9.3% for vitamin B12 deficiency anemia. Additionally, 151 (74%) of children had received medical treatment, and 53 (26%) of them had received no medical treatment for anemia. All details related to sociodemographic characteristics of parents and children are given in Table 1.

### 3.2. Results of P-CPQ

The answers to two general questions which were asked to parents before the Turkish version of the P-CPQ, “How would you rate the health of your child’s teeth, lips, jaws and mouth?” and “How much is your child’s overall well-being affected by the condition of his/her teeth, lips, jaws or mouth?” are shown in Table 2.

More than half of parents (56.3%) responded “*fair*” for the health of their children’s teeth, lips, jaws and mouth. Similarly, the child’s overall well-being was stated as being affected “*a lot*” by the condition of their child’s teeth, lips, jaws or mouth by half of parents (49.5%). 

Besides the abovementioned two general questions, the descriptive statistical analyses (the values of mean, standard error, and percentage) of each item in all four subscales with general impacts are given in Table 3. 

The subscale scores for oral symptoms (OS), functional limitations (FL), emotional well-being (EWB), and social well-being (SWB) domains were calculated as 12 ± 0.413, 10.36 ± 0.385, 7.75 ± 0.412 and 7.13 ± 0.513, respectively. Moreover, the P-CPQ total score (TS), which was determined by summing up the scores of all 31 items, was 36 ± 1.36. The highest percentages of impacts were reported as “*often*” in the oral symptoms subscale for the items of “*pain in teeth and mouth/45.1%*”, “*bleeding gums/38.2%*”, “*mouth sores/37.3%*”, “*bad breath/40.4%*”, and “*food caught between teeth/39.7%*” by parents. Furthermore, few “*don’t know (DK)*” responses, which were accepted as “*missing*”, were recorded. Only 11 parents ticked DK as an answer in one item of oral symptoms and three items of emotional and social well-being.

The results of association between overall and subscale scores of the P-CPQ with demographic factors are given in Table 4, and more than one statistically significant comparisons were obtained in the subscale scores regarding demographic factors. Firstly, the scores of functional limitations between parental relationship were detected to be significant (*p* = 0.032). Mothers (10.01 ± 0.44) reported less functional limitation compared to fathers (12.05 ± 0.749). Another statistically-significant difference was detected between the educational level of parents in the oral symptoms subscale (*p* = 0.002). The parents with higher educational levels reported a higher impact in comparison to parents with lower education levels (8.38 ± 0.866, 11.43 ± 0.419, respectively). When the responses for medical treatment for anemia (yes/no) were analyzed, all differences in each subscale and total score were observed as significant (*p* = 0.006 for OS, *p* < 0.001 for FL, *p* = 0.008 for EWB, *p* = 0.03 for SWB, and *p* < 0.001 for TS). To elaborate on this further, the parents whose children take medical treatment for anemia reported a larger effect on children in all subscale scores. Another significant difference was observed for the sensitive analysis conducted between health professionals and unemployed groups in the oral symptom subscale. Health professional groups responded that their children were more affected in the oral symptom subscale compared to those of unemployed parents (*p* = 0.04). 

### 3.3. Results of ECOHIS

The responses to each item in the ECOHIS questionnaire are represented in Table 5 by giving detailed descriptive statistical analyses (the values of mean, standard error and percentage). First of all, a large number of children (81.5%) reported occasional or more frequent oral/dental pain. Secondly, the subscale scores were determined for child symptoms (2.25 ± 0.067), child function (6.8 ± 0.22), child psychology (3.87 ± 0.128), self-image and social interaction (1.74 ± 0.063), parental distress (3.82 ± 0.143), and family function (3.5 ± 0.121). The values of child impact, family impact and total score were calculated according to responses to each item (16.3 ± 0.486, 7.32 ± 0.242, 23.6 ± 0.701, respectively). Moreover, a detailed analysis on descriptive statistics of the ECOHIS further showed that several children (160, 79.2%) had difficulty drinking and that 152 (76.7%) children had trouble eating food occasionally or more frequently. According to family impact section responses, the majority of parents reported that they felt upset (142/70.6%) and guilty (138/69.3%) occasionally or more frequently. Likewise, 73.1% of parents stated that they take time off from work, and 69% of parents reported that their child’s dental procedures occasionally or more frequently had a financial impact on the family.

The association between overall and subscale scores of the ECOHIS and demographic data is given in Table 6. 

The statistically-significant results were observed in groups of educational level and medical treatment for anemia. The most dramatic results were seen in the medical treatment group for parents with children taking medical treatment for anemia, who reported a much larger impact on the child and family and larger total scores in comparison to parents with children not taking medical treatment (*p* < 0.001 for all comparisons). Regarding parental education level, the results for primary–high-school graduated parents for child impact and total score were lower than those for graduate–postgraduate parents (*p* < 0.016, *p* < 0.025 respectively). An additional pairwise comparison regarding employment status and occupations of parents revealed that the comparison between non-health professional and unemployed parents had a statistically significant value in the total score (*p* = 0.028).

### 3.4. Internal Consistency Evaluation

The examination of the internal consistency of each questionnaire was performed by determining Cronbach’s alpha. Cronbach’s alpha values of the ECOHIS were 0.918, 0.859, and 0.946 for the sections of child impact, family impact and total score, respectively. 

In terms of the P-CPQ, Cronbach’s alpha values ranged from 0.844 to 0.941. Cronbach’s alpha for the oral symptoms, functional limitation, emotional well-being, social well-being, and total scale was calculated as 0.932, 0.844, 0.853, 0.916 and 0.941, respectively.

All details for Cronbach’s alpha evaluation are given in Table 7.

## 4. Discussion

The objective of the present study was to evaluate oral-health-related quality of life (OHRQoL) for children and for parents regarding the dental effects of anemia. Within the study, Turkish versions of the P-CPQ and ECOHIS were preferred to measure the OHRQoL of parents and children. Assessing the parental opinion of children’s OHRQoL is important from different perspectives, since the child’s health and their parents’/caregivers’ opinion are correlated and reveal the treatment needs of the child. Moreover, the reports of parents/caregivers provide more comprehensive analysis of the child’s OHRQoL [28]. Hence, a strength of the current study was that it followed the Turkish version of the P-CPQ that was adapted cross-culturally, with a good internal consistency and reliability of subscales regarding parents [27]. The impact of anemia on OHRQoL was measured using the reliable and valid Turkish version of the ECOHIS questionnaire to expose the effects of anemia-related dental caries and to create a strategy for oral health improvement for children and their families [26]. To the best of our knowledge, there are no similar studies regarding OHRQoL of anemia-related dental caries in the literature.

Turning back to the present findings of the P-CPQ, it is clear that more than half of the parents stated their children’s oral health and well-being was affected substantially by the presence of anemia. The perceptions of parents of oral QoL could be associated with oral symptoms of their child, including the complaints of pain in the teeth and mouth, bad breath, and food caught between teeth. Hence, the results of questions which were answered prior to the P-CPQ showed greater agreement with the study of Gültekin et al. [27]. When the parental relationship is evaluated within the subscale scores, fathers reported a more negative impact on the functional limitations of their child compared to mothers. The mentioned result has been clarified by better mother–child communication, which allows mothers to more reliably monitor their child’s difficulties during drinking and eating. Even though the results of the current study could not be directly compared with the study of Fernandes et al. [29]. due to different points of investigation, their featured results concerning parental monitorization of their child were partially consistent with the present results. Moreover, families with higher education level (graduate–postgraduate) reported greater negative impact on the oral symptoms subscale of their child in comparison with primary–high school-educated families. A possible reason for this is the better knowledge of the anemia–oral effect relationship of families with higher education level. 

According to the latest report of the World Health Organization (WHO), it was estimated that 42% of children aged less than five years are anemic. Iron deficiency, including deficiencies in folate and vitamins B12 and A, is considered the main cause of anemia. In addition, hemoglobinopathies and infectious diseases, such as malaria, tuberculosis, and HIV, and parasitic infections, are reported as important etiological factors of anemia [30]. Iron deficiency is the predominant etiological factor of anemia, showing numerous oral manifestations (e.g., angular cheilitis, angular stomatitis, erythematous stomatitis, atrophic glossitis) across countries [31]. Similarly, iron-deficiency anemia was detected at a high level in the current study. Besides the beforementioned oral symptoms of anemia, black-stain formations on tooth surfaces are also among the most common oral manifestations due to iron supplementation during anemia treatment [32]. Strikingly, however, medical treatment for anemia significantly affected all subscales of the P-CPQ consisting of oral symptoms, functional limitations, emotional well-being, social well-being, and overall well-being in the present study. Therefore, the negative impact of medical treatment for anemia on children’s well-being in all attitudes may be associated with the pharmacological side effects of anemia syrups, especially iron supplements, in our study.

When the results of the ECOHIS questionnaire were evaluated, there was a correlation between anemia and oral-health impact. Generally, the most negative impacts were revealed on the functions of children during drinking/eating, and their symptoms related to oral/dental pain in a similar vein to the studies of Farsi et al. [33] and Hashim et al. [34]. The ECOHIS findings also were comparatively evaluated from the standpoint of each sociodemographic factor. Even though a weak and non-significant relationship was observed between parental relationship and all subscale scores of the ECOHIS, a strong correlation was observed between education level of parents and the child-impact section. Moreover, similar to the answers to the P-CPQ, parents with graduate—postgraduate education level reported that anemia created greater negative effects on their child. Besides the similarity in educational levels of two different questionnaires, medical treatment for anemia showed powerful adverse impacts on all subscales of the ECOHIS along with the P-CPQ. 

The reliability and Internal consistency of each questionnaire were evaluated by using Cronbach’s alpha. Cronbach’s alpha values for both questionnaires were satisfactory for the current study. For the ECOHIS, Cronbach’s alpha values were close to those of Turkish version of the ECOHIS (0.981, 0.859, 0.946 for child impact section, family impact, and total score, respectively) [26]. For the P-CPQ, Cronbach’s alpha estimation showed high internal consistency reliability and parallelism with the original Turkish version (0.932, 0.844, 0.853, 0.916, 0.941 for oral symptoms, functional well-being, emotional well-being, social well-being, and total score, respectively [27]. 

Lastly, the lack of data on age of anemia diagnosis was the major limitation in the present research. However, importantly, the age at which anemia was diagnosed could not be the exact age of disease onset. It was therefore possible that data related to age of anemia diagnosis were not always reliable. 

The clinical significance of this work is that understanding the adverse relationship between anemia-related ECC and quality of life of children and parents may lead to the design of cost-effective and approachable international preventive programs, especially for infants and preschool children.

The most important findings of the study are as follows:Anemia-related dental caries leads to negative impacts on the quality of life of children and families.A large number of children (81.5%) reported occasional or more frequent oral/dental pain.More than half of parents (56.3%) responded “*fair*” for the health of their children’s teeth, lips, jaws, and mouth. Similarly, children’s overall well-being was stated as being affected “*a lot*” by the condition of their child’s teeth, lips, jaws or mouth by half of parents (49.5%).The age of anemia diagnosis is a very significant issue in analyzing the main mechanism of anemia in ECC development.Further investigations must be designed to determine the exact oral adverse effects of anemia related to ECC.

## 5. Conclusions

Within the limitations of the present study, it may be concluded that anemia-related dental caries has a highly negative impact on the quality of life of children and parents, according to both questionnaires. Therefore, it could be possible to prioritize children with high scores for preventive procedures and timely dental treatments. However, further investigations must be developed to understand the exact oral effects of anemia by further evaluation of each subscale. 

## Figures and Tables

**Table 1 medicina-59-00521-t001:** Sociodemographic characteristics of parents and children.

Characteristics	Number (*n*)	Percentage (%)
Demographic characteristics of parents		
Parental Relationship		
Mother	146	71.6
Father	58	28.4
Age (Years)		
21–30	26	12.7
31–40	104	51.0
>40	74	36.3
Educational Level		
Primary–high school	42	20.7
Graduate–postgraduate	161	79.3
Employment Status and Occupation		
Non-health professional	103	50.5
Health professional	78	38.2
Unemployed	23	11.3
Marital Status		
Married	176	86.3
Single	28	13.7
Family Income (Monthly/EUR) *		
333 (minimum wage)	45	22.3
334–555	57	28.2
>555	100	49.5
Demographic characteristics of children		
Gender		
Male	94	46.1
Female	110	53.9
Anemia type		
Iron deficiency anemia	111	54.4
25-hydroxyvitamin D deficiency anemia	60	29.4
Vitamin B12 deficiency anemia	19	9.3
Other	13	6.4
Medical treatment for anemia		
Yes	151	74.0
No	53	26.0

* Monthly family income measured in TRY (Turkish lira, EUR 1 = TRY 18).

**Table 2 medicina-59-00521-t002:** Child’s oral health and well-being from Parental Caregiver Perception Questionnaire (P-CPQ).

Health of Your Child’s Teeth, Lips, Jaws and Mouth	Number (*n*)	Percentage (%)
Excellent	11	5.4
Very good	14	6.9
Good	52	25.5
Fair	115	56.4
Poor	12	5.9
Child’s oral well-being affected by the condition of his/her teeth, lips, jaws or mouth		
Not at all	14	6.9
Very little	20	9.8
Some	51	25
A lot	101	49.5
Very much	18	8.8

**Table 3 medicina-59-00521-t003:** Descriptive statistics and summary of Parental Caregiver Perception Questionnaire (P-CPQ) responses (*n* = 204).

			*n* (%)					
Impacts	Mean	SE *	Never: 0	Once/Twice: 1	Sometimes: 2	Often: 3	Every Day/Almost Every Day: 4	Missing
Oral symptoms subscale (6 items)								
Pain in teeth and mouth	1.917	0.082	36(17.6)	41(20.1)	33(16.2)	92(45.1)	2(1)	0
Bleeding gums	1.667	0.087	56(27.5)	34(16.7)	36(17.6)	78(38.2)	0(0)	0
Mouth sores	1.735	0.0845	46(22.5)	42(20.6)	38(18.6)	76(37.3)	2(1)	0
Bad breath	2.005	0.0767	27(13.3)	35(17.2)	55(27.1)	82(40.4)	4(2)	1
Food stuck to roof of mouth	1.461	0.07	51(25)	31(15.2)	101(49.5)	19(9.3)	2(1)	0
Food caught between teeth	1.995	0.075	26(12.7)	36(17.6)	58(28.4)	81(39.7)	3(1.5)	0
General	12	0.413						
Functional limitation subscale (8 items)								
Difficulty chewing firm food	1.922	0.0823	38(18.6)	30(14.7)	51(25)	80(39.2)	5(2.5)	0
Breathing through mouth	0.956	0.0787	102(50)	36(17.6)	43(21.1)	19(9.3)	4(2)	0
Trouble sleeping	1.495	0.0686	47(23)	30(14.7)	110(53.9)	13(6.4)	4(2)	0
Unclear speech	0.701	0.071	127(62.3)	25(12.3)	41(20.1)	8(3.9)	3(1.5)	0
Slow eating	1.451	0.0619	40(19.6)	44(21.6)	110(53.9)	8(3.9)	2(1)	0
Difficulty drinking/eating hot/cold foods	1.265	0.0636	40(19.6)	92(45.1)	52(25.5)	18(8.8)	2(1)	0
Difficulty eating preferred foods	1.564	0.0649	39(19.1)	31(15.2)	117(57.4)	14(6.9)	3(1.5)	0
Restricted diet	1.235	0.0627	49(24)	70(34.3)	75(36.8)	8(3.9)	2(1)	0
General	10.6	0.385						
Emotional well-being subscale (7 items)								
Upset	2	0.0789	33(16.2)	29(14.2)	48(23.5)	93(45.6)	1(0.5)	0
Irritable/frustrated	0.971	0.0839	111(54.4)	25(12.3)	32(15.7)	35(17.2)	1(0.5)	0
Anxious/fearful	1.851	0.0893	49(24.3)	31(15.3)	24(11.9)	97(48)	1(0.5)	2
Worried about being different from other people	0.716	0.0757	131(64.2)	24(11.8)	26(12.7)	22(10.8)	1(0.5)	0
Worried he or she is less attractive than others	0.775	0.0792	127(62.3)	25(12.3)	25(12.3)	25(12.3)	2(1)	0
Shy (embarrassed)	0.833	0.0803	120(59.1)	27(13.3)	28(13.8)	26(12.8)	2(1)	1
Worried about having few friends	0.522	0.0703	150(73.9)	19(9.4)	19(9.4)	11(5.4)	4(2)	1
General	7.75	0.412						
Social well-being subscale (10 items)								
Missed school	1.328	0.0679	50(24.5)	58(28.4)	77(37.7)	17(8.3)	2(1)	0
Had hard time paying attention in school	0.824	0.0804	122(59.8)	26(12.7)	29(14.2)	24(11.8)	3(1.5)	0
Not wanting to speak/read aloud in class	0.74	0.0773	128(62.7)	24(11.8)	35(17.2)	11(5.4)	6(2.9)	0
Not wanting to talk other children	0.611	0.0674	134(66)	27(13.3)	30(14.8)	11(5.4)	1(0.5)	1
Avoiding smiling when around other children	1.118	0.0683	65(32)	67(33)	55(27.1)	14(6.9)	2(1)	1
Teased/called names by other children	0.348	0.0578	163(79.9)	24(11.8)	7(3.4)	7(3.4)	3(1.5)	0
Left out by other children	0.324	0.0507	162(79.4)	25(12.3)	10(4.9)	7(3.4)	0	0
Not wanting/unable to be with other children	0.539	0.0615	137(67.2)	33(16.2)	26(12.7)	7(3.4)	1(0.5)	0
Not wanting/unable to take part in activities	0.529	0.0686	149(73)	19(9.3)	21(10.3)	13(6.4)	2(1)	0
Asked by other children about the condition	0.705	0.0763	127(63.5)	29(14.5)	23(11.5)	18(9)	3(1.5)	4
General	7.13	0.523						
Total Score	36	1.36						

* Standard error.

**Table 4 medicina-59-00521-t004:** Association between overall and subscale scores of Parental Caregiver Perception Questionnaire (P-CPQ) with demographic factors.

	Subscale Scores																									
	Oral Symptoms (OS)				Functional Limitations (FL)				Emotional Well-Being (EWB)				Social Well-Being (SWB)				Total Score (TS)									
	Mean	Median	*p* value	SE *	Mean	Median	*p* value	SE *	Mean	Median	*p* value	SE *	Mean	Median	*p* value	SE *	Mean	Median	*p* value	SE *						
Parental Relationship																										
Mother	10.47	11.00	0.237	0.488	10.01	11.00	0.032	0.440	1.71	6.00	0.628	0.502	6.87	3.00	0.222	0.624	34.84	37.00	0.143	1.639						
Father	11.52	12.00		0.775	12.05	12.00		0.749	7.83	6.00		0.716	7.75	4.00		0.962	39.07	38.00		2.380						
																						*p* value				
Age (Years)																					Pairwise comparisons	OS	FL	EWB	SWB	TS
21–30	8.04	8.00		1.050	9.88	9.50		1.310	7.12	4.50		1.250	8.60	5.00		1.610	33.30	32.00		4.440	21–30/31–40	0.002	0.402	0.445	0.905	0.613
31–40	12.00	13.00		0.547	11.30	11.00		0.453	8.07	6.00		0.574	6.82	3.00		0.689	37.90	37.00		1.690	21–30/>40	0.314	0.993	0.799	0.684	0.992
>40	10.00	10.00		0.714	9.88	12.00		0.711	7.53	6.00		0.673	7.04	3.50		0.908	34.40	37.00		2.440	31–40/>40	0.064	0.703	0.854	0.784	0.574
Educational Level																										
Primary–high school	8.38	8.50	0.002	0.866	11.02	12.00	0.532	0.939	8.17	7.00	0.390	0.817	8.54	6.00	0.177	1.167	35.98	38.00	0.696	2.906						
Graduate–postgraduate	11.43	12.00		0.419	10.40	11.00		0.416	7.59	6.00		0.476	6.76	3.00		0.584	35.93	37.00		1.543						
																						*p* value				
Employment Status and Occupation																					Pairwise comparisons	OS	FL	EWB	SWB	TS
Non-health professional	10.70	11.00		0.579	10.40	12.00		0.555	8.17	6.00		0.607	7.94	4.00		0.784	37.00	38.00		1.980	Non-health professional/health professional	0.711	0.872	0.801	0.34	0.766
Health professional	11.60	13.00		0.670	11.60	11.00		0.558	7.52	6.00		0.661	6.23	3.00		0.794	35.70	37.00		2.110	Non-health professional/unemployed	0.153	0.916	0.599	0.69	0.795
Unemployed	8.30	8.00		1.140	11.40	11.00		1.410	6.61	6.00		0.963	6.52	5.00		1.380	32.80	37.00		4.070	Health professional/unemployed	0.04	0.928	0.881	0.996	0.928

Marital Status																										
Married	10.87	12.00	0.661	0.437	10.32	11.00	0.076	0.395	7.67	6.00	0.570	0.443	6.89	3.00	0.188	0.553	35.48	37.00	0.216	1.452						
Single	10.14	10.00		1.244	12.29	12.00		1.281	8.21	6.50		1.128	8.63	5.00		1.564	39.57	39.00		3.818						
																						*p* value				
Family Income (Monthly/EUR) *																					Pairwise comparisons	OS	FL	EWB	SWB	TS
333 (minimum wage)	11.50	13.00		0.951	10.90	12.00		0.782	7.38	6.00		0.677	6.95	4.00		1.030	36.50	38.00		2.440	333/334–555	0.76	0.691	0.891	0.863	0.779
334–555	10.80	11.00		0.832	10.40	11.00		0.720	6.93	6.00		0.736	7.02	3.00		1.050	35.00	37.00		2.560	333/>555	0.317	0.895	0.798	0.984	0.701
>555	10.40	11.00		0.549	10.60	11.00		0.573	8.46	6.00		0.659	7.32	3.00		0.756	36.50	37.00		2.100	334–555/>555	0.828	0.942	0.507	0.904	0.987

Gender of child																										
Male	9.91	10.00	0.090	0.606	10.15	11.00	0.198	0.545	7.57	6.00	0.849	0.592	7.59	4.00	0.493	0.775	35.03	37.00	0.358	1.986						
Female	11.30	12.00		0.565	10.92	12.00		0.553	7.94	6.00		0.589	6.83	3.00		0.721	36.84	38.00		1.913						
																						*p* value				
Anemia type																					Pairwise comparisons	OS	FL	EWB	SWB	TS
Iron deficiency anemia (IDA)	11.40	12.00		0.551	10.40	11.00		0.467	7.00	6.00		0.518	5.90	3.00		0.633	34.40	37.00		1.680	IDA/DA	0.908	0.145	0.041	0.003	0.038
25-hydroxyvitamin D deficiency anemia (DA)	10.80	12.00		0.734	11.80	12.00		0.740	9.95	7.00		0.826	10.20	8.50		1.080	42.60	40.00		2.620	IDA/B12A	0.956	0.551	0.714	0.599	0.401
Vitamin B12 deficiency anemia (B12A)	10.50	11.00		0.370	12.00	12.00		1.510	8.21	7.00		1.280	8.39	4.50		1.740	39.60	40.00		4.390	DA/B12A	0.999	0.999	0.824	0.836	0.974
Medical treatment for anemia																										
Yes	11.52	12.00	0.006	0.452	11.50	12.00	<0.001	0.426	8.48	6.00	0.008	0.499	7.82	4.00	0.030	0.640	39.10	38.00	<0.001	1.540						
No	8.64	8.00		0.878	8.13	8.00		0.762	5.61	6.00		0.611	5.17	3.00		0.812	27.30	30.00		2.470						

* Standard error.

**Table 5 medicina-59-00521-t005:** Descriptive statistics and summary of Early Childhood Oral Health Impact Scale Responses (ECOHIS) (*n* = 204).

			*n* (%)					
Impacts	Mean	SE *	Never/0	Hardly Ever/1	Occasionally/2	Often/3	Very Often/4	Don’t Know (Missing)/0
Child Impacts	16.3	0.486						
Child symptoms subscale (1 item)								
Oral/Dental Pain	2.245	0.0678	15(7.5)	22(11)	68(34)	89(44.5)	6(3)	4(0)
General Score	2.25	0.0678						
Child function subscale (4 items)								
Difficulty Drinking	2.262	0.0714	18(8.9)	24(11.9)	53(26.2)	101(50)	6(3)	2(0)
Difficulty Eating	2.242	0.0738	18(9.1)	28(14.1)	46(23.2)	100(50.5)	6(3)	6(0)
Difficulty Pronouncing Words	0.851	0.0768	109(50.4)	38(18.8)	35(17.3)	16(7.9)	4(2)	2(0)
Missed Preschool or School	1.575	0.0651	34(17)	41(20)	103(51.5)	20(10)	2(1)	4(0)
General Score	6.8	0.22						
Child psychology subscale (2 items)								
Trouble Sleeping	2.005	0.0774	27(13.4)	37(18.4)	47(23.4)	88(43.8)	2(1)	3(0)
Irritable or Frustrated	1.94	0.0667	16(8)	30(15.1)	119(59.8)	18(9)	16(8)	5(0)
General Score	3.87	0.128						
Self-image and social interaction subscales (2 items)								
Avoided Smiling or Laughing	1.739	0.0635	27(13.3)	29(14.3)	123(66.6)	18(8.9)	6(3)	1(0)
Avoided Talking	1.7	0.07	31(15.3)	38(18.7)	104(51.2)	21(10.3)	9(4.4)	1(0)
General Score	1.74	0.0635						
Family Impacts	7.32	0.242						
Parental distress subscale (2 items)								
Being upset	1.801	0.0691	25(12.4)	34(16.9)	109(54.2)	22(10.9)	11(5.5)	3(0)
Felt guilty	2.095	0.0851	31(15.6)	30(15.1)	38(19.1)	89(44.7)	11(5.5)	5(0)
General Score	3.82	0.143						
Family function subscale (2 items)								
Time off from Work	1.846	0.0662	22(10.9)	32(15.9)	110(54.7)	29(14.4)	8(4)	3(0)
Financial Impact	1.715	0.0674	29(14.5)	33(16.5)	111(55.5)	20(10)	7(3.5)	4(0)
General Score	3.5	0.121						
Total Score	23.6	0.701						

* Standard error.

**Table 6 medicina-59-00521-t006:** Association between overall and subscales scores of Early Childhood Oral Health Impact Scale Responses (ECOHIS) and demographic data.

	Subscale Scores															
	Child Impact				Family Impact				Total Score							
	Mean	Median	*p* value	SE *	Mean	Median	*p* value	SE *	Mean	Median	*p* value	SE *				
Parental Relationship																
Mother	16.14	19.00	0.860	0.613	7.20	9.00	0.685	0.300	23.34	27.50	0.857	0.885				
Father	16.67	19.50		0.742	7.62	9.00		0.393	24.29	2800.00		1.061				
														*p* value		
Age (Years)													Pairwise comparisons	Child Impact	Family Impact	TS
21–30	12.90	12.00		1.300	6.12	6.00		0.662	19.00	19.50		1.850	21–30/31–40	<0.001	0.009	<0.001
31–40	17.70	20.00		0.609	8.10	9.00		0.299	25.80	29.00		0.869	21–30/>40	0.141	0.661	0.217
>40	15.50	16.00		0.878	6.65	8.00		0.441	22.10	24.50		1.280	31–40/>40	0.102	0.012	0.028
Educational Level																
Primary–high school	14.74	15.00	0.016	0.813	6.64	7.00	0.096	0.416	21.38	22.00	0.025	1.150				
Graduate–postgraduate	16.75	20.00		0.573	7.45	9.00		0.283	24.20	29.00		0.830				
														*p* value		
Employment Status and Occupation													Pairwise comparisons	Child Impact	Family Impact	TS
Non-health professional	17.10	20.00		0.676	7.66	9.00		0.314	24.70	29.00		0.967	Non-health professional/health professional	0.680	0.633	0.674
Health professional	16.10	20.00		0.815	7.18	9.00		0.425	23.30	29.00		1.180	Non-health professional/unemployed	0.014	0.141	0.028
Unemployed	13.30	15.00		1.210	6.26	6.00		0.728	19.70	21.00		1.780	Health professional/unemployed	0.062	0.503	0.128

Marital Status																
Married	16.20	19.00	0.939	7.040	7.23	9.00	0.563	0.268	23.40	28.50	0.900	0.774				
Single	16.80	19.00		6.400	7.89	8.50		0.530	23.60	27.00		1.570				
														*p* value		
Family Income (Monthly/EUR) *													Pairwise comparisons	Child Impact	Family Impact	TS
333 (minimum wage)	15.90	20.00		0.926	7.730	9.00		0.461	23.60	29.00		1.310	333/334–555	0.794	0.448	0.765
334–555	15.30	19.00		0.908	7.000	9.00		0.426	22.30	27.00		1.290	333/>555	0.730	0.780	0.943
>555	17.00	19.00		0.734	7.310	8.00		0.377	24.30	26.50		1.070	334–555/>555	0.368	0.883	0.517

Gender of child																
Male	15.24	16.00	0.052	0.758	7.05	8.00	0.472	0.385	22.30	25.00	0.112	1.107				
Female	17.06	20.00		0.638	7.49	9.00		0.316	24.55	29.00		0.912				
														*p* value		
Anemia type													Pairwise comparisons	Child Impact	Family Impact	TS
Iron deficiency anemia (IDA)	16.60	20.00		0.628	7.39	9.00		0.317	24.00	29.00		0.906	IDA/B12A	0.999	0.834	0.999
25-hydroxyvitamin D deficiency anemia (DA)	16.80	18.50		0.963	7.70	9.00		0.452	24.50	26.50		1.380	IDA/DA	0.649	0.991	1.000
Vitamin B12 deficiency anemia (B12A)	17.50	19.00		1.360	8.16	9.00		0.735	25.60	27.00		1.840	B12A/D2A	1.000	0.942	0.995


Medical treatment for anemia																
Yes	17.50	20.00	<0.001	0.538	7.91	9.00	<0.001	0.274	25.50	29.00	<0.001	0.771				
No	12.70	13.00		0.918	5.64	6.00		0.439	18.40	18.00		1.330				

* Standard error.

**Table 7 medicina-59-00521-t007:** Cronbach’s alpha results.

Impacts	Validity Percentage (%)	Number of Items	Cronbach’s α
P-CPQ impacts			
Oral symptoms subscale	99.5	6	0.932
Functional limitation subscale	100	8	0.844
Emotional well-being subscale	98	7	0.853
Social well-being subscale	97.1	10	0.916
Total score	94.5	31	0.941
ECOHIS impacts			
Impacts			
Child impacts	92.5	9	0.918
Family impacts	95.6	4	0.859
Total score	89.7	13	0.946

## Data Availability

The data presented in this study are openly available.

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
