# Peer review of "The Impact of Anemia-Related Early Childhood Caries on Parents’ and Children’s Quality of Life"

_medicina, 2023, doi:10.3390/medicina59030521_

Round 1
Reviewer 1 Report
"The Impact of Anaemia Related Early Childhood Caries on Parents and Children Quality of Life" well organized article, it is very important subject which we should know.
Reviewer 2 Report
This is a well designed research study. I guess it will improve and provide better information to the readers and the scientists with making following minor revisions.
The text should be re-checked into native English with grammatical corrections.
Before going forward, it will be better if the authors could add information about the clinical significance at the discussion section.
Reviewer 3 Report
Dear authors,
Congradulations on your work. I suugest some revisisons as well as an extensive language editing
1. Language and grammar:
The reviewer strongly reccomneds a professional check on grammar particulary.
Following is a minor example of the corrections needed:
Today, oral diseases are well known for their effects on not only daily life but also quality of life (QoL). sSince oral health exhibites the main physiological, social and psychological characterictics that are fundamental for the quality of life [1,2]. In this sense, the evaluation of oral health related to quality of life (OHRQoL) has to be come into prominence as well as oral health. The existence of dental disesase, treatment experience and oral health issues canmay create unfavourablye effects on the quality of daily life of children and alsoand their parents [3-5]. Dental caries specially early childhood caries (ECC) are is considered as a public health concern by having impact on child’ s and parent’s life quality from multiple aspects such as pain, eating disorders, sleeping problems, take time off from school, social embrassment for children and financial problems related with treatment fees, time off work for parents [6-8]. Herein, evaluating and measuring the OHRQoL to help displaying the priority of care and the interpretation of treatment outcomes have gained popularity [9]. Eventhough various studies have mentioned about the multifactorial etiology of ECC including behavioral, socioeconomic, biological and environmental factors, the relationship between anaemia (particularly associated with malnutrition, iron deficiency and vitamin D deficiency) and ECC has recently been highlighted. According to the recent studies, it has been shown that there is a correlation between ECC and anaemia which is defined as the number of red blood cells or their oxygen carrying capacity of blood being below the normal intervals. [10-13]. Particularly, the possible mechanism of iron deficiency anaemia on the development of dental caries is explained by the potential inhibitorry effect of iron on cariogenic microorganisms [14]
2. Inclusion and exclusion criteria
The reviewer strongly advice to re-organize this section by giving more detail. Are the patients with accompanying dieases inclueded (i.e anemia and other systemical disorders)When werethe patiemts diagnosed (i.e bfore age 3 or after? The time of diagnosis may affect the outcome of tested quesitionnaire)
3. Th aouthors may give more detailed information about the DK answers. Were they in the lower educated group?
4. The discussion section is relativelt better organised. The reviwer congradulates the aouthors for the good work here. On the other hand, it may be helpful for the reader that if the authors underline the most important findings as bulltes
Sincerely
Reviewer 4 Report
The research is focused on the impact of the anemia related early childhood caries. In my opinion the topic is original and of a high interest for the readers. The methods are accurate as two independent stages were realized. The number of children participating is relevant. The conclusions are consistent with the evidence and arguments presented and they address the main question posed. The references are appropriate, their number can be increased. In my opinion the article should be accepted.Author Response
Please see the attachment.

Reviewer 5 Report
Thank you for the opportunity to provide a review about this article. An interesting topic relates things that are still rarely explored regarding early childhood caries with anemia, but there are several things that need input, including:
1. In the introduction, the reasons regarding the theoretical basis for the link between anemia and ECC were not discussed so that it did not provide a research gap regarding the need for this research.
2. In the method section, please describe the measuring instruments in this study. ECOHIS and PCP-Q experience transadaptation? what about the validity and reliability? is it tested? because what I know is that the two measuring instruments are not native to the author's area.
Then how is the process of distributing this measuring instrument to the participants? please describe
3. For subjects who experience ECC, as we all know, there are three types of ECC. So, is this ECC grouped based on these types? and how to measure it
4. how many investigators examined the clinical condition? is there an inter-investigator agreement? how to determine it?
